# Sustainable Internet of Vehicles System: A Task Offloading Strategy Based on Improved Genetic Algorithm

**Kun Wang** [1,*]**, Xiaofeng Wang** [2] **and Xuan Liu** [3]

1   College of Physics and Electronic Engineering, Shanxi University, Taiyuan 030006, China
2   School of Electric Power, Civil Engineering and Architecture, Shanxi University, Taiyuan 030032, China
3   Shanxi Electric Power Company Maintenance Branch, State Grid, Taiyuan 030000, China
*   Correspondence: eekunwang@126.com

**Abstract:** "Smart transportation" promotes urban sustainable development. The Internet of Vehicles (IoV) refers to a network with huge interaction, which comprises location, speed, route information, and other information about vehicles. To address the problems that the existing task scheduling models and strategies are mostly single and the reasonable allocation of tasks is not considered in these strategies, leading to the low completion rate of unloading, a task offloading with improved genetic algorithm (GA) is proposed. At first, with division in communication and calculation models, a system utility function maximization model is objectively conducted. The problem is solved by improved GA to obtain the scheme of optimal task offloading. As GA, in the traditional sense, inclines to a local optimum, the model herein introduces a Halton sequence for uniform initial population distribution. Additionally, the authors also adapt improved GA for the problem model and global optimal solution guarantee, thus improving the rate of task completion. Finally, the proposed method is proven through empirical study in view of scenario building. The experimental demonstration of the proposed strategy based on the built scenario shows that the task calculation completion rate is not less than 75%, and when the vehicle terminal is 70, the high-priority task completion rate also reaches 90%, which can realize reasonable allocation of computing resources and ensure the successful unloading of tasks.

**Keywords:** IoV; improved GA; task offloading; sustainable development; adaptive dynamic weight; system utility function maximization

## 1. Introduction

"Smart transportation" promotes urban sustainable development [1]. Internet of Vehicles sends the status of vehicles through sensor technology and wireless communication—modern intelligent technology for information processing—thus realizing the smart management of traffic, as in examples such as smart traffic service information decisions and control over vehicles [2]. IoV generates economic effects at a large scale. Improvement in IoV would improve scenarios for new application formations, thus driving 5th-generation mobile networks (5G) with scale effects and deployment as systems for intelligent transportation development, as well as intelligent connected vehicles. Vehicles have been endowed with more capabilities for computing, storage, sensing, and communication. While these advancements provide users with a more realistic and convenient experience, these data must be processed. Many new in-vehicle applications, including augmented reality, virtual reality, appearance recognition, and pattern recognition, also increase the scale of in-vehicle data processing exponentially. This results in higher requirements for the transmission of data and network processing capacities. The ability to process massive quantities of data accurately and quickly is not only related to user experience, but is also an important guarantee of the safety of drivers and passengers. Thus, the vehicle network has higher requirements for data transmission and processing [3]. However, the massive

data processing capability of a mobile device's own central processing unit (CPU) alone is not enough to meet the latency and power consumption required by service applications. For solving insufficiency in device computing resources, the concept of cloud computing has been proposed and widely-used [4]. Cloud computing can maximize the elastic utilization of computing resources by virtue of its efficient centralized and shared resources. Since cloud servers are deployed in a more centralized location, stored data can be more easily protected, and lifecycle management is simpler than for distributed systems [5]. As cloud servers and devices are a long distance apart, and their signal bandwidth is limited, the latency of service applications cannot be reliably guaranteed. As mobile devices and Internet of Things (IoT) devices have grown explosively, their limitations have become more prominent [6].

With the continuous development of the IoT and 5th-generation (5G) communication technology, on-board mobile cloud computing (MCC) has begun to transform into mobile edge computing (MEC) [7,8]; thus, the concept of MEC came into being. MEC emphasizes the importance of computing resources in cloud servers being closer to the user side. Deploying MEC servers on the mobile processing terminal facilitates the transmission and processing of data. Task offloading is a key technology of MEC, which entails the uploading of some or all tasks generated on the vehicle to the edge computing server or cloud server. Vehicle task uploading to its computing edge server reduces the pressure on the vehicle's local computing, but also avoids the queuing delay of the elastic compute service, reduces task energy and delay consumption, and improves the completion rate of the task. Much research has been undertaken on task offloading in MEC; however, some problems must still be addressed, such as vehicle mobility management, computing resource allocation, and task offloading decision-making under limited computing resources.

The current MEC offloading strategies are all concentrated on small mobile terminals, which have limited energy consumption, slow movement, and often do not span the service scope of a single MEC in a short time. However, in the IoV scenario, the vehicle terminal moves quickly and interacts with multiple MEC servers in a short time. As such, determining the appropriate MEC server on which the vehicle terminal should offload the task to enable the fastest solution to the computing task is a topic worth studying. The multiservice of the IoV is not considered in calculation offloading. According to the application requirements of standard 3GPP, the delay of a small number of services is 500 ms, and the delay limit of 1 s is only approximate. However, these services are vehicle safety services, which are closely related to personal safety, and should therefore have higher processing priority. Thus, the various services of the IoV have different performance requirements with respect to communication. For example, the delay standards and reliability requirements of onboard safe services are different. Thus, an offloading strategy should meet the delay requirements of multiple IoV services, and the computing tasks of higher priority should be handled first. Based on improved GA, a task offloading method is proposed. Experiments showed that the vehicle computing resource utilization, in terminal end and MEC, are improved by optimizing GA allocation. Major innovations of this paper are as follows: Using a division between communication and calculation models, a system utility function maximization model is objectively conducted. The problem is solved by improved GA to obtain the scheme of optimal task offloading, thus reducing computing task unloading to the MEC server which has heavy loading, and increasing light-loaded MEC server-based computing task unloading, achieving balance in loading effects.

The Section 2 is about existing task scheduling models and strategies; the Section 3 establishes the system with which actual complex road conditions are modeled; the Section 4 introduces the strategy of task offloading with an improved GA; the Section 5 details the experiment designed to verify the performance of the strategy proposed by the paper; and the Section 6 presents the conclusions of this paper.

## 2. Related Works

At present, some research on MEC has been performed in various countries, and the task offloading problem of edge computing has been a primary concern of scholars. Research has mostly started by focusing on the task offloading problems: the computation of task offloading optimization, the joint optimization of multiple resources, or the mobility of edge computing [9]. Among these objectives, the goal of optimization mainly entails reducing the overall delay of tasks and energy consumption, and optimizing overall benefit by combining the delay and energy consumption [10]. For example, in [11], based on reinforcement learning computing, they implemented a task offloading strategy in the IoV edge computing architecture. The vehicle network model was constructed through collected data to ensure that the vehicle network task offloading was rational. For user cost function minimization, they used a double-layer deep Q-network for solving real-time changes in network state due to the movement of users. However, the processing efficiency of task offloading needs to be improved. Wang et al. (2020) [12] proposed a task offloading method with guidance from meta-reinforcement learning. The offloading strategy of a custom sequence-to-sequence neural network was combined to gradient update and samples at small amounts to quickly adapt to new environments, which effectively improved the processing efficiency of computing tasks.

Resource optimization has primarily been achieved for computing resources, transmission resources, and caching. For example, [13] designed a new network architecture after in-depth research on vehicle self-organizing network architecture in the vehicle network and the typical application of vehicle networking. The architecture featured greater data throughput, lower latency, higher security, and massive connectivity. This method achieved optimal resource occupation; however, the objective of optimization was not sufficiently considered. Xue et al. (2021) [14] proposed a vehicle-assisted MEC, which meant that on-board computing tasks could be offloaded to MEC servers and vehicle edge nodes. By establishing a differentiated pricing model and dynamic incentive model based on different resource states, the optimal offloading strategy and pricing scheme were obtained. The combination of gradient-based iterative algorithms for resource allocation effectively improved rationality with regard to resources of computing, but it showed insufficiency for objective function optimization of computing task offloading. By optimizing particle swarm, Dai et al. (2020) [15] proposed a strategy of offloading data computing tasks for mobile medical applications. The proposed algorithm was evaluated against the local computation method as a baseline method through extensive simulations, and the results showed that the proposed task assignment scheme had good feasibility and achieved high completion efficiency. However, its handling of multi-target mobile computing task offloading needed to be improved.

The mobility of users in MEC also represents a hot topic in current research. For example, in [16], vehicle computing task offloading was defined as a fatal problem with multi-armed bandit and online algorithms, which were newly used for realizing the decision-making of node selection distribution. With the edge nodes of context information, infinite exploration space was converted into a finite exploration space, and simulation results verified its effectiveness. Wang et al. (2020) [17] proposed a joint task offloading and transfer strategy in reinforcement-learning-based MEC networks for maximum system gain. Considering the time-varying computing tasks and resource conditions, the Markov decision process was used to solve the mixed integer nonlinear programming problem, which effectively improved MEC computing task offloading efficiency. Li et al. (2021) [18] studied a UAV-assisted multi-task MEC network considering the requirements of time-sensitive tasks. In addition to satisfying different task requirements, they effectively reduced the energy consumption of IoT devices in total. With MEC, Li et al. (2021) proposed a control framework for the SDN IoV with three layers [19]. Then, a strategy of controlled placement was used for obtaining the controller's optimal location through Louvain algorithm, within the index of load balance and buffer size. Liwang et al. (2022) [20] used a resource trading approach with novel futures for Internet of Vehicles (EC-IoV), with capability

for edge computing through use of a forward contract for promoting negotiations with resource trading between an MEC server (seller) and a vehicle (buyer) in a given future term. However, in the IoV scenario, the vehicle terminal moves quickly and interacts with multiple MEC servers in a short period. Thus, determining the appropriate MEC server on which the vehicle terminal should offload the task to enable the fastest solution to the computing task is a topic worth studying. To accelerate execution of tasks with an MEC server, Nguyen et al. (2021) [21] suggested a scheme of task computing through collaboration with intentional encouragement from an MEC server with rich resources nearby, thus participating in this scheme. Results of their study found that the scheme had positive effects on data redundancy migration. In [22], regarding these issues, the technology of blockchain was used to ensure reliable data transmission and interaction. By taking vehicle task computation offloading decision with optimization into consideration, decisions were cached, as were the nodes of offloaded consensus quantity. Thus, there was a decrease in interval, size of block, energy consumption, and computation overheads. Using a decentralized framework for MEC, Anwar et al. (2022) [23] proposed a dynamic path with shortest distance based on a matrix, and an algorithm with the function of dynamic multipath searching was selected through a boundary discovery with autonomous network with necessary connections to node block benchmarks. The experimental results found network QoS efficiency was better than the centralized method. The above methods demonstrated the sufficient consideration of mobility, but task processing delay and energy consumption represent areas that still need to be strengthened.

As most IoV task scheduling models and strategies have this problem, which is simple, rate of offloading completion is low for reasonable task allocation. Thus, improved GA-based task offloading for the IoV is proposed.

## 3. System Model

### 3.1. Network Topology

Figure 1 shows the MEC system. Roadside units (RSUs) are uniformly deployed beside the road, and an MEC server-equipped RSU is used to process the tasks of vehicles on the road; there are no differences among the RSUs. Meanwhile, indivisible computing task consideration is $s_i = \{D_i, C_i, T_i^{max}\}$, where $D_i$ is the current task data size $i$, $C_i$ is the number of CPU revolutions of task $i$, and $T_i^{max}$ shows the maximum tolerable task delay for task $i$. Computing a task to its edge server can be achieved through localization or uploading; thus, all vehicles are maintained within the same level of driving by default.

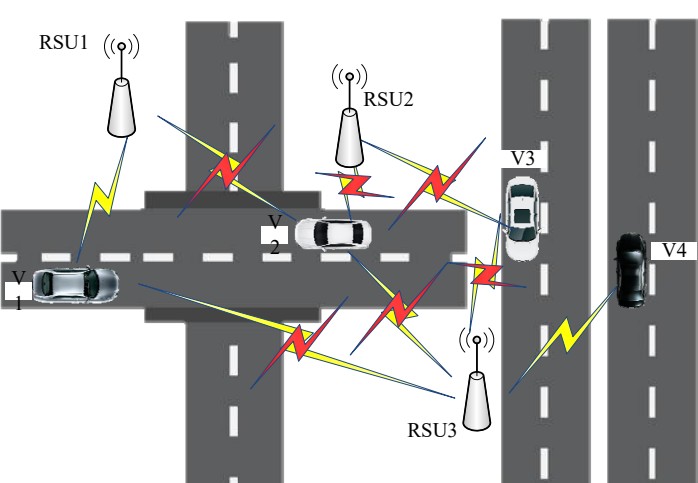

**Figure 1.** IoV architecture based on multi-edge computing.

The goal of the system is the co-optimization of delay and energy consumption. With certain system optimizations, delay and energy consumption optimization will oppose each other. The pursuit of low energy consumption will result in high latency, while the

pursuit of low latency will inevitably result in greater energy consumption. Thus, overhead $\psi_i$ is designed, which considers the delay and energy consumption in a unified manner, and the calculation is as follows:

$$\psi_i = \lambda_i T_i + (1 - \lambda_i) E_i \tag{1}$$

In the formula, $T_i$ and $E_i$ represent the current task-based delay and energy consumption; and $\lambda_i$ is the weight factor, $\lambda_i \in [0, 1]$. The size of the weight factor is vital to optimize the offloading scheme. When the value of $\lambda_i$ is large, it greatly affects system delay; otherwise, it will affect energy consumption greatly. For solving this, an adaptive weight $\lambda_i$ is designed, which can dynamically adjust weight delay and energy consumption according to the situation of the task itself ($C_i$ and $T_i^{\max}$). If $C_i$ of the current task is large and $T_i^{\max}$ is small, then the weight of the task's delay will be greater. The solution process of adaptive weight is as follows:

$$\lambda_i^* = \xi \frac{C_i}{\sum_{i=1}^{N} C_i} + (1 - \xi) \frac{\sum_{i=1}^{N} T_i^{\max}}{T_i^{\max}} \tag{2}$$

where $\xi$ is the weighting factor, $0 < \xi < 1$. The principle to be followed for the selection of $\xi$: the value of the task that is sensitive to the delay should be reduced, so $\xi = 0.5$ is taken. Finally, the obtained weights are normalized:

$$\lambda_i = \frac{\lambda_i^* - \min(\lambda^*)}{\max(\lambda^*) - \min(\lambda^*)} \tag{3}$$

For offloading computing, there are four possible scenarios: (1) during task offloading computing, no base station handover; (2) base station handover during computing task upload; (3) base station handover during computing task processing; and (4) base station is switched during computing task result return. The computational tasks considered in this paper are indivisible. Thus, for the second scenario, when some computing tasks are uploaded to the base where a handover occurs, some data offloading for original base is invalid, and the computing tasks need to be re-uploaded. For the third and fourth scenarios, processing of computing task is achieved from the edge computing server located in original base station, and through base station data linkage, the computing task results are migrated to a new station, which is responsible for the results transmission of computing task to the vehicles. As data quantity in the computing task results is extremely small, the data migration delay can be ignored.

*3.2. Communication Model*

The wireless link data upload rate $V_{up}$ in the IoV is calculated as follows:

$$V_{up} = B_{up} \log_2 \left( 1 + \frac{P_i d_1^{-\delta} h^2}{N_0} \right) \tag{4}$$

where $B_{up}$ shows the width of upload channel band, $d_1^{-\delta}$ shows the loss of path for vehicle and RSU, $P_i$ represents vehicle power, $d_1$ shows the gap between the vehicle and RSU, $\delta$ refers to loss of path, $h$ represents upload link factor with channel fading, and $N_0$ refers to power of white Gaussian noise.

Vehicle speed in this scenario is constantly unidirectional, represented by $Q_i$ which, as displayed through $v_i$ (vehicle mobility), facilitates the distance $d_l$ from vehicle to RSU coverage area center along with time; thus, variation law is as shown:

$$d_l(t) = \sqrt{d_\Delta^2 + \left( \frac{z}{2} - v_i t \right)^2} \tag{5}$$

where $d_\Delta$ shows how far driving level of vehicle is from RSU, and $z$ represents RSU coverage, which refers to the distance from the vehicle to the RSU coverage area center. To make the issue simple, rate of upload on average, $\overline{V_{up}}$, shows the rate of data transmission of vehicle offloading computing tasks to the MEC server during the offloading process of the IoV, and $t_{stay}$ represents the time from the initial position of the vehicle to when it leaves the coverage of the RSUs:

$$\overline{V_{up}} = \frac{\int_0^{t_{stay}} V_{up}(t)dt}{t_{stay}} \tag{6}$$

*3.3. Computational Model*

The processing of computing tasks is divided into two parts: transmission and calculation of data. By taking task $i$, which performs local computing, only its computing delay is taken into consideration, rather than transmission delay. Calculation of delay in local task procession $t_{loc}$ is as follows:

$$T_i = t_{loc} = g_i / f^{loc} \tag{7}$$

where $f^{loc}$ is vehicle terminal computing capability, and $g_i$ represents the calculation amount of task $i$.

For tasks that need to be offloaded, there are two cases: local server offloading and other server offloading. MEC servers where direct vehicle communication is within the link of wireless scope are defined as local servers. Within current coverage area, the vehicle offloads its tasks of computing to the MEC server. When results are calculated by the server, they are returned to the vehicle immediately. The total delay that constitutes a computing task is mainly composed of two parts: computing task upload delay and MEC server processing delay, the latter of which considers offloads to the MEC server, computing delay, and transmission delay. $t_{ij}^{mec}$ represents the processing delay of offloading task $i$ to server $j$, $e_{ij}^{mec}$ represents computing resource allocation through server $j$ to task $i$, and $v_{ij}$ shows how far it is from the vehicle to which the computing task belongs and server $j$. The wireless transfer rate is calculated as follows:

$$t_{ij}^{mec} = \frac{g_i}{e_{ij}^{mec}}, t_{ij}^{trans} = \frac{D_i}{v_{ij}}$$
$$T_i = t_{ij}^{mec} + t_{ij}^{trans} \tag{8}$$

The above equations assume vehicle terminal has offloaded computing tasks to other MEC servers for being processed in the current range. MEC servers are connected by wired links such as optical fibers, and it is assumed that computing tasks' average latency transmission with wired link $l$ is $t_w$. Here, calculation of the delay for task processing is as follows:

$$t_{ij} = \frac{g_i}{e_{ij}^{mec}} + t_{ij}^{trans} + 2\tau t_w \tag{9}$$

where $\tau$ represents the number of wired link hops made by computing tasks to be offloaded to MEC servers in other ranges.

To guarantee the completion of task without interruption in the limited time, the computing task is required to be completed in advance of the vehicle leaving the MEC cell scope [24]. Therefore, with offloaded local server, this satisfies the criteria below:

$$\begin{cases} t_i^{stay} = z_i / v_i \\ \frac{g_i}{e_{ij}^{mec}} + t_{ij}^{trans} \le \min\left[t^{\max}, t_i^{stay}\right] \end{cases} \tag{10}$$

where $z_i$ represents the distance from the vehicle of the MEC server within its coverage area.

If other servers are offloaded, the following should be satisfied:

$$\frac{g_i}{e_{ij}^{mec}} + t_{ij}^{trans} + 2\tau t_w \leq \min\left[t^{\max}, t_i^{stay}\right] \tag{11}$$

Thus, required computing resources for task computing completion are deduced:

$$e_{ij}^{mec} \geq \frac{g_i}{\min\left[t_i^{\max}, t_i^{stay}\right] - t_{ij}^{trans}}$$
$$e_{ij}^{mec} \geq \frac{g_i}{\min\left[t_i^{\max}, t_i^{stay}\right] - t_{ij}^{trans} - 2\tau t_w} \tag{12}$$

The total computing resources $E_j$ requested by all offload tasks for computing to the $mec_j$ are as shown:

$$E_j = \sum_{i=1}^{N} \sum_{x_i=j} e_{ij}^{mec} \tag{13}$$

### 3.4. Problem Definition

When determining the priority of computing tasks, three factors are primarily considered: data quantity for computing task, the computing resources occupied by messages, and the requirement for the delay deadline. The deadline is more important than the amount of message data and the computing resources occupied by messages, and the computing resources occupied by messages are more important than the amount of message data. Thus, in the AHP model, at the highest weight, requirement for deadline has divisions in priority. Firstly, the factors at the same level are compared in pairs, and the AHP matrix $A = \left(a_{jk}\right)_{n \times n}$ is constructed:

$$a_{jk} = \begin{cases} \frac{1}{a_{kj}} = n, n = \{1, 2, \cdots, 9\}, j \neq k \\ 1, j = k \end{cases} \tag{14}$$

When computing the vector of comparison matrix weights, the most common summation method is used. The calculation formula corresponding to the weight is shown:

$$U_k^j = \frac{\sum_{j=1}^{n} a_{jk}}{\sum_{j=1}^{n} \sum_{k=1}^{n} a_{kj}} \tag{15}$$

Then, the composition matrix $\Delta$ corresponding to the weights of all the messages is as follows:

$$\Delta = \begin{bmatrix} u_1^1 u_1^2 u_1^3 \\ u_2^1 u_2^2 u_2^3 \\ \vdots \\ u_k^1 u_k^2 u_k^3 \end{bmatrix} \tag{16}$$

Next, the eigenvalues corresponding to the weights are obtained according to the layer-side analysis matrix, which is represented by $\Lambda$; that is, $\Lambda = [\gamma_1, \gamma_2, \gamma_3]^T$, where, by definition:

$$\gamma_b = \frac{1}{b} \sum_{j=1}^{b} \frac{\sum_{j=1}^{b} a_{jk}}{\sum_{k=1}^{b} a_{jk} u_k} \tag{17}$$

where $b$ represents the number of influencing factors involved in the process of making decisions.

Finally, the priority $\varpi$ of each computing task is obtained, and each element in $\varpi$ represents the priority of computing tasks, which is calculated as follows:

$$\varpi = \Delta\Lambda = \begin{bmatrix} u_1^1 u_1^2 u_1^3 \\ u_2^1 u_2^2 u_2^3 \\ \vdots \\ u_k^1 u_k^2 u_k^3 \end{bmatrix} \cdot [\gamma_1, \gamma_2, \gamma_3]^T = \begin{pmatrix} \sum_{b=1}^{3} u_1^b \gamma_b \\ \sum_{b=1}^{3} u_2^b \gamma_b \\ \vdots \\ \sum_{b=1}^{3} u_k^b \gamma_b \end{pmatrix} \tag{18}$$

The purpose of this solution is to prioritize the processing of computing tasks of higher priority while increasing the number of completed computing tasks. Thus, the system utility function $P$ is defined as follows:

$$P = \sum_{i=1}^{N} \varpi_k / \vartheta \tag{19}$$

where $\vartheta$ represents the total weight value of the computing task.

In this scheme, the range of priority weight $\varpi$ of a task is equally divided into five parts. The computing task whose $\varpi$ value belongs to the largest range is defined as the $\varpi$ computing task with the highest priority and weight, where $R_\varpi$ represents the extreme difference in the $\varpi$ values of each computing task, and $\varpi$ is used to represent the vehicle safety computing task.

The final computational model is as follows:

$$\begin{aligned} & \max(P) \\ & s.t. \mathrm{C1} : y_i \in \{0, 1\}, \forall i \in N \\ & \quad\ \ \mathrm{C2} : e_j \le E_j, j = \{1, 2, \cdots, J\} \end{aligned} \tag{20}$$

C1 represents the offloading decision; 0 represents local computing; 1 represents the offload value of MEC for computing; C2 represents the computing resources required which do not exceed the computing resources that can be provided.

## 4. Task Offloading Strategy Based on Improved GA

### 4.1. GA Selection

In solving complex problems, algorithms are often used to reduce problem difficulty. Algorithms are mainly divided into traditional algorithms and artificial intelligence algorithms [25]. Traditional optimization algorithms deal with relatively simple problems, such as certain linear programming problems. Their structures are simple, clear, and easy to solve. Intelligent optimization algorithms can deal with a variety of difficult problems. So long as the parameters of the problem meet the input requirements of the intelligent algorithm, the problem can be solved. In general, intelligent algorithms can deal with more types of problems, and are more general with respect to problem solving.

The theory of GA surrounds an intelligent algorithm derived from natural selection and genetics in biology. This algorithm first generates a population of a fixed size, and then crosses and mutates the population by setting a genetic operator similar to those of biological genetics. General individuals are eliminated via the fitness function, and the most common individual is finally selected as the solution to the problem. The solution flow of the GA is shown in Figure 2.

Compared with traditional algorithms, GA has a faster convergence speed, and the population is randomly generated, meaning that precision is greater. The algorithm is simple and easy to implement, and it can obtain multiple approximate solutions at the same time. Although the GA is superior to traditional algorithms with respect to solving complex problems, the GA also has certain shortcomings [26,27]. The algorithm is difficult to design and needs to be encoded and decoded. Moreover, the selection of genetic operators randomly affects the results, and the parallel mechanism is not effectively utilized.

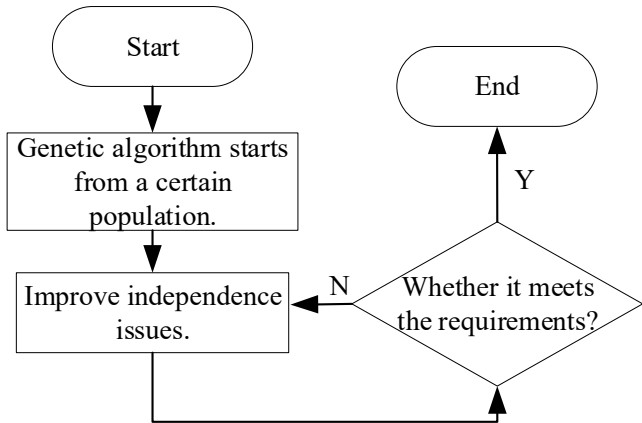

**Figure 2.** Solution flow of GA.

*4.2. GA Improvement*

Standard GAs tend to fall into local optimal solutions during the solution process. After multiple iterations, the characteristics of the population remain basically unchanged, and different populations are generated only through crossover and mutation operations. Thus, an improved GA based on adaptive change of the genetic operator is designed in this study. The adaptive GA is based on the basic GA, and realizes the balance between randomness and searching through adaptive adjustment of the genetic operator. Specifically, when the individual values in the population are relatively concentrated and the diversity is poor, the mutation probability and crossover probability are increased. When the population fitness is scattered and the diversity is high, the mutation probability and crossover probability are appropriately reduced. The improved algorithm is divided into two parts, micro and macro:

(1)  Microscopic genetic strategy: From the perspective of algorithm parameter setting, the setting of the genetic operator and population size (and its influence on the solution result) are discussed;

(2)  Macro genetic strategy: This primarily originates from the process of the GA, and involves the optimization of the algorithm process or the introduction of other intelligent algorithms to improve the ability of legacy algorithms to solve the objective function.

The selection of the GA genetic operator has a great influence on the convergence accuracy and speed of the GA. Genetic operators in traditional GAs give a fixed value. When the mutation probability is too large, the population diversity increases. However, this model makes it easy to fall into a local optimum, wherein the overall optimum solution cannot be obtained. When the mutation rate is too low, the diversity of the population decreases and no new individuals are produced. For different optimization objectives, if the value of the mutation operator is determined through repeated experiments, the process is complicated and difficult to operate. In order to retain the most individuals in the population and ensure that new individuals can be introduced, it is necessary to improve the mutation probability in the traditional GA.

In the late stages of algorithm iteration, difference in the population is small, and the competitiveness of excellent individuals becomes obviously insufficient, making it easy for the algorithm to eliminate the optimal individual and fall into a local optimum. The genetic operators in the GA, including crossover and mutation probability, have a great influence on the problem results, which is embodied by the diversity of a population and the search for the overall optimal solution. Thus, when setting the parameters of the GA, the probability of crossover and mutation should not be too large or too small. This paper therefore designs an adaptive adjustment based on the probability of crossover and mutation.

*4.3. Solving Steps of Improved GA*

The first step of the GA is to generate the initial population, which is required in order to obtain information on the target. The use of Halton sequences can make the individual differences between populations smaller, which cancels randomness to reduce population differences. In this paper, the Halton sequence is used to generate the initial population with less difference. The Halton sequence is introduced to randomly generate 100 points. In comparing this with the initial population generated by a standard GA, the difference in the initial number of Halton sequences generated was small. It was also more uniform with respect to the axis distribution number, which greatly improves the quality of the initial population and provides a good foundation for the subsequent optimization of the algorithm.

For the design of the fitness function, considering that the goal of GA optimization is to maximize utility function, utility function in the problem is used as the fitness function to evaluate an individual's pros and cons. The constraints in the optimization problem are guaranteed during the initialization and selection operations.

Regarding the chromosome encoding method, a combination of binary encoding and floating-point encoding is used. The binary coding method adopts a 0/1 coding method, and 0/1 on each bit can represent an information state. Thus, if the binary string is long enough, it can cover all the state information. Binary encoding is easier to operate, but an inevitable mapping error will occur between the continuous and discrete values. Due to precision requirements, the length of string cannot be too short. However, longer strings will complicate the decoding process and cause the dramatic expansion of the search space. Floating-point encoding represents individual characteristics through a floating-point number, which is very helpful for representing numbers with a larger range. Thus, combined with the optimization objective function, a combination of binary encoding and floating-point encoding is used to find the optimal allocation strategy. Binary encoding is used to encode offloading strategies, and floating-point encoding is used to calculate the resource allocation strategy. The chromosome code of each individual can therefore be expressed as follows:

$$L_i^{mec} = [L_1, \cdots, L_i, \cdots, L_N] \tag{21}$$

where $L_i = [y_i, e_i]^T$ is the combination of the user $i$ offloading strategy and computing resource allocation strategy. If $y_i = 0$, then $L_i = [0,0]^T$.

For selection operators, the random tournament selection method is a good choice because of its low computational complexity and good individual selectivity. Two individuals are randomly selected each time, and the better individual is kept until the number of selected individuals reaches the initial population size. If the best individual is ignored in the selection operation, the selected next-generation individuals are sorted to find the individual with the worst fitness value and replace it with the best individual. In order to increase the individual richness of population, the feasible solutions are selected first, and then the infeasible solutions near the constraint boundary are selected because, after the next iteration, infeasible solutions near the boundary will likely become feasible.

For the design of the crossover operator, due to the different encoding methods of offloading decision set $Y$ and the computing resource allocation strategy set $F$, they are operated independently with probability $P_c$. For set $X$, this section uses the uniform crossover operator, which can speed up convergence and prevent trapping in local extrema. For the method of reorganization used in the set, the expression of the intersection operator is as below:

$$\begin{cases} \phi_i(child1) = (1-\varepsilon)\phi_i(parent1) + \varepsilon\phi_i(parent2) \\ \phi_i(child2) = \varepsilon\phi_i(parent2) + (1-\varepsilon)\phi_i(parent2) \end{cases} \tag{22}$$

where $\varepsilon$ is a random variable whose value range is between 0 and 1; and $\phi_i$ is the computing resource allocation strategy of the server.

Finally, regarding mutation operations, the operations on sets $Y$ and $F$ are also different. The 0/1 replacement for each $y_i$ is performed with probability $P_m$; under these constraints, a random variable is randomly added or subtracted to each $\phi_i$. The probability is also $P_m$.

Through the above analysis and design of each genetic step in the improved GA, the task offloading process based on the improved GA can be obtained, and its specific description is shown in Algorithm 1.

---

**Algorithm 1.** Pseudo-code of task offloading based on improved GA

---

**Input**
$N_u$, $K$, $P_c$, $P_m$, $T$
**Begin**
1. **Initialization**: Under the constraint of the optimization problem, initialize the population in a random way, and the number of individuals is $K$.
2. Calculate the fitness value of the individual, select the largest individual as $L_{best}$, and its fitness value is $Q_{best}$.
3. Set the number of iterations $T$.
4. **For** $t = 1:T$ **do**

    Two individuals are randomly selected, and the crossover operation is carried out with
    probability $P_c$. The methods of uniform crossover and recombination are adopted for
    $y$ and $f$, respectively.
    Select individuals from parents and offspring with probability $P_m$ for mutation operation.
    Calculate the fitness value of each new individual and divide it into feasible individuals and
    infeasible individuals.
    The method of random tournament is used to select the best individual.

5. Compare the optimal individual $L'_{best}$ of iteration $t$ with the historical optimal individual $L_{best}$. If $L'_{best}$ is better than $L_{best}$, let $L_{best} = L'_{best}$ and update $Q_{best}$.
6. **End For**
7. Output: $L_{best}$, $Q_{best}$
**End**

---

The time complexity of the task offloading algorithm based on the GA depends on the time complexity of the selection operation, crossover operation, and mutation operation. The time complexity of the selected operation is $O(N_{iter} \times N_p \times N_{ts})$. The time complexity of the crossover operation and mutation operation is $O(N_{iter} \times N_p)$. Thus, the overall time complexity of the algorithm is $O(N_{iter} \times N_p \times N_{ts})$.

## 5. Experiment and Analysis

In the experiment, the coverage of each MEC server was set to 100 m~120 m, the number of vehicle terminals was 10~70, and each vehicle offloaded 5~15 concurrent computing tasks. All the vehicles were randomly assigned a speed between 50 km/h and 100 km/h and drove at a constant speed, and the initial positions of the vehicles were randomly-distributed on the road. The detailed simulation parameters of the edge car networking system are shown in Table 1.

**Table 1.** Simulation parameters of edge vehicle networking system.

| Parameters | Value |
|---|---|
| Average waiting delay of wired link/ms | 5~20 |
| Task upload rate/(kb/s) | 1000 |
| MEC computing power/(cycles/s) | $5 \times 10^7$~$9 \times 10^7$ |
| Computing capacity of vehicle terminal/(cycles/s) | $5 \times 106$ |
| Vertical distance from community center to road/m | 10~40 |

The computing task parameters and the simulation parameters of the improved GA are shown in Table 2.

**Table 2.** Simulation parameters of computing tasks and improved GA.

| Parameters | Value |
| --- | --- |
| Task time limit/ms | 250~600 |
| Task calculation amount/(cycles/bit) | 20~80 |
| Task data size/KB | 50~120 |
| Population number | 50 |
| Mutation probability | 0.8 |
| Crossover probability | 0.1 |
| Maximum number of iterations | 200 |

*5.1. Convergence Performance of Algorithms for Different Computation Tasks*

In order to demonstrate the convergence of the improved GA, it was compared with the traditional GA, the results of which are illustrated in Figure 3.

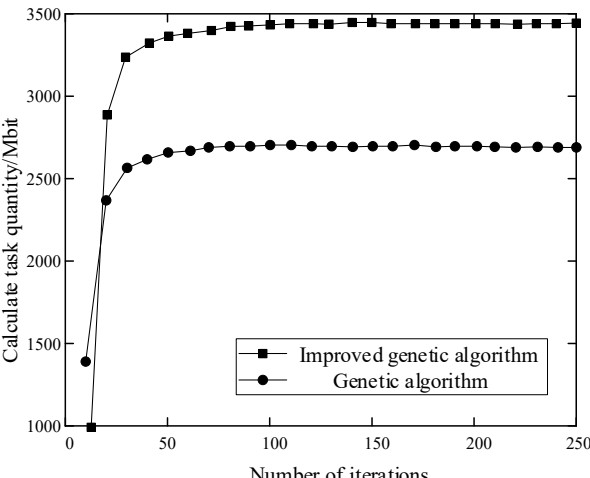

**Figure 3.** Convergence performance of GA before and after improvement.

Figure 3 shows that the proposed improved GA executed more tasks within the same number of iterations, and when the number of iterations exceeded 85, the number of computation tasks tended to converge to 3440 Mbit.

*5.2. Relationship between the Number of Offloading Tasks and the Total Number of Network Tasks*

In order to observe the advantages of the proposed strategy more intuitively, three schemes were set up for comparison: Local offoad, all computing tasks are processed locally; Tocal offoad, all computing tasks are offloaded to MEC servers for processing; Proposed strategy, according to the proposed method, all tasks are first classified and sorted, and then the tasks are offloaded to different devices for processing. The relationship between the number of tasks offloaded to the MEC server and the total number of tasks is shown in Figure 4.

As can be observed from Figure 4, 'Local offoad' places all the tasks locally in the vehicle for processing, so there is no task offloading; hat is, the number of offloaded tasks is 0. 'Tocal offoad' offloads all the tasks to the MEC server for processing. When the number of tasks is greater than 24, the computing resources of the MEC server are exhausted, and the remaining tasks can only be selected for processing. 'Proposed strategy' adopts the proposed strategy to offload tasks, and 20 computing tasks are selected to be offloaded to MEC servers for processing. A total of 10 computing tasks are selected to be processed locally in the vehicle, and the rational allocation of tasks greatly improves resource utilization efficiency.

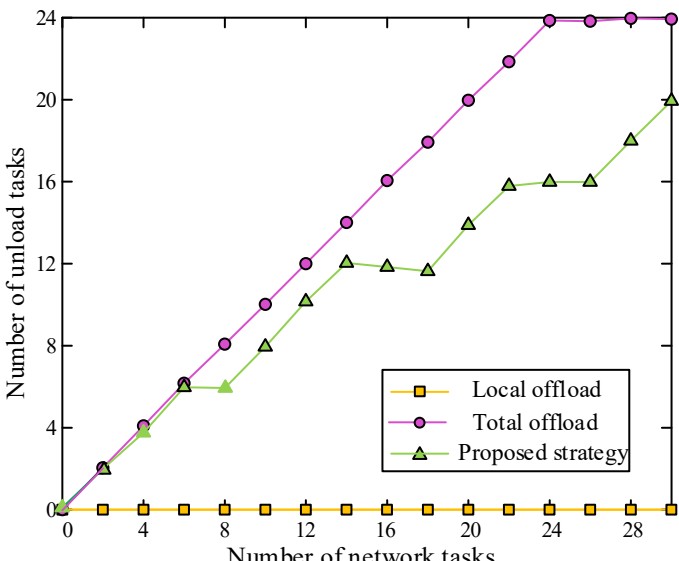

**Figure 4.** Relationship between the number of offloading tasks and the total number of network tasks.

*5.3. Influence of Load Imbalance on Task Completion Rate*

When the number of fixed on-board terminals is 45 and the total computing resources of *M* MEC servers are $6 \times 10^8$ cycles/s, the effect of an uneven load of MEC servers on the task completion rate is as shown in Figure 5. The MEC server load unevenness factor $\Theta$ is calculated as follows:

$$\Theta = \lg\left(\frac{1}{1000000M}\sum_{i=1}^{M}\left(E_i - \overline{E}\right)^2\right) \tag{23}$$

where $\overline{E}$ is the mean value of the computing resources used by the servers.

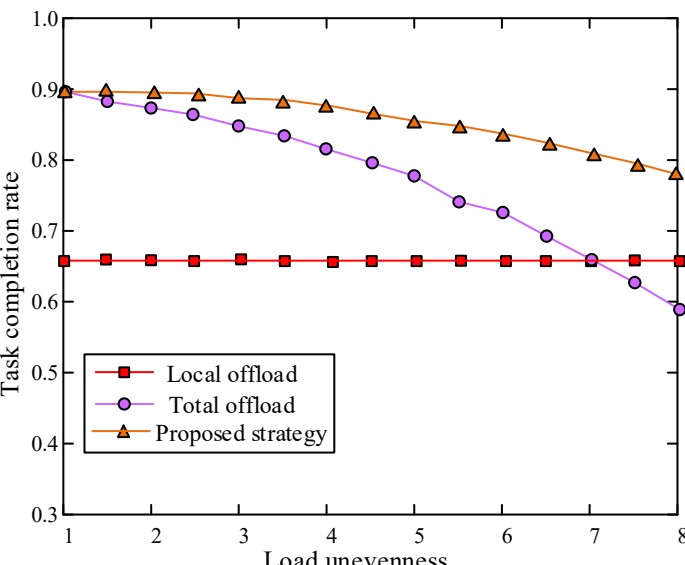

**Figure 5.** Relationship between load imbalance and task completion rate.

It can be seen from Figure 5 that the overall offloading strategy cannot guarantee the completion rate of computing tasks due to the deepening of uneven load of the MEC server, and the completion rate is less than 70%. When the proposed strategy uses improved genetic algorithm to iteratively optimize the decision-making scheme, it reduces the offloading of computing tasks to the heavily-loaded MEC server. The offloading of computing tasks is added to the MEC server with light load to achieve the effect of load balancing. Compared

with other strategies, this strategy is less affected by the uneven load of MEC servers. When the load unevenness is 8, its task completion rate is close to 80%. The full offloading strategy can only offload computing tasks to an MEC server in the current range. Therefore, when the load is extremely uneven, the MEC server with scarce computing resources can only process a small part of computing tasks, and the processing of a large number of computing tasks fails. Because the total computing resources of the MEC server is a fixed value, it increases with the increase of computing tasks. All offloading policies are more affected by an uneven load on the MEC server.

### 5.4. Completion Rate of Computing Tasks under Different Offloading Schemes

The task completion rates corresponding to the four offloading strategies are shown in Figure 6, where the number of vehicles increases gradually.

It can be observed from Figure 6 that, as the number of vehicles per unit time continues to increase, the task completion rate under all four unloading strategies decreases. When there are few computing tasks generated by vehicles, the proposed strategy and the strategies in ((Wang K, et al. (2020)); (Xue J, et al. (2021)); (Wang D, et al. (2020))) [11,14,17] can effectively offload computing tasks. However, with the increase in the number of computing tasks per unit time, the strategy in ((Wang D, et al. (2020)); (Xue J, et al. (2021))) [14,17] gradually becomes less effective than the strategy in (Wang K, et al. (2020)) [11]. This is because the computing resources of a single MEC server are roughly the same as the computing resources of 10 in-vehicle terminals, and the increasing number of vehicles causes the local computing power to gradually exceed MEC. These two strategies cause a large number of computing task processing failures because too many computing tasks are offloaded to the MEC server.

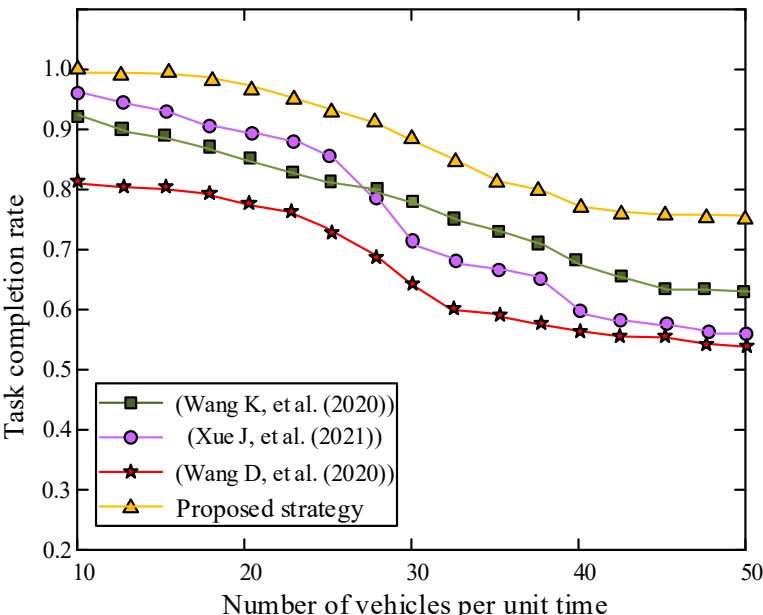

**Figure 6.** Computing task completion rates of different offloading strategies (Wang K, et al. (2020)); (Xue J, et al. (2021)); (Wang D, et al. (2020)) [11,14,17].

### 5.5. Completion Quantity of Vehicle-Mounted Safety Computing Tasks with Different Offloading Schemes

Figure 7 shows the number of completed on-board safety computing tasks with different offloading schemes.

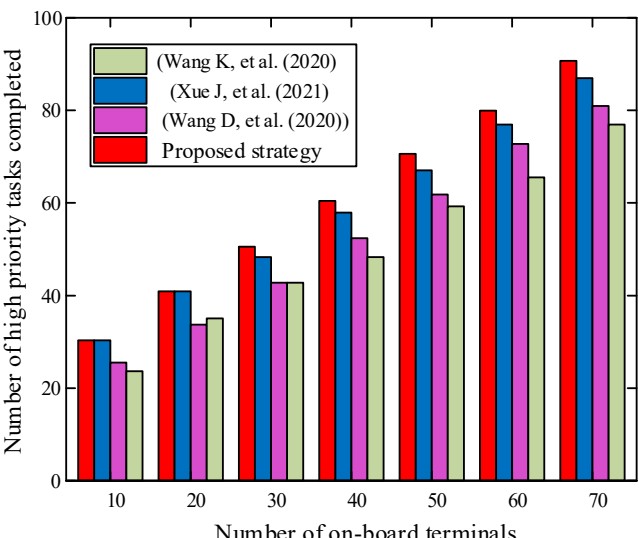

**Figure 7.** Number of on-board safety computing tasks completed with different offloading schemes (Wang K, et al. (2020)); (Xue J, et al. (2021)); (Wang D, et al. (2020)) [11,14,17].

As can be observed from Figure 7, when the number of on-board terminals is 70, the completion rate of high-priority tasks is close to 90%. In addition, the average delay of the proposed strategy increases slowly. When the number of on-board terminals is 70, the average delay is about 21.2 ms. When the proposed strategy uses the improved GA to iteratively optimize the decision-making scheme, it reduces the offloading of computing tasks to the heavily-loaded MEC server. The offloading of computing tasks is added to the MEC server with a light load to achieve the effect of load balancing. Compared with other strategies, this strategy is less affected by uneven MEC server load. The strategy in (Xue J, et al. (2021)) [14] performs at about the same level as the proposed strategy with fewer vehicles. Because the number of vehicles is small at this time and the generated computing tasks are insufficient, the MEC server can process all the offloaded computing tasks. The strategy in ((Wang K, et al. (2020)); (Wang D, et al. (2020))) [11,17] can utilize the computing resources of the vehicle terminal itself and the computing resources of the MEC server to offload tasks. Thus, some in-vehicle safety computing tasks are processed by local computing.

## 6. Conclusions

In recent years, the IoV has attracted widespread attention, and various applications have gradually emerged, improving road safety, traffic efficiency, and driving comfort. However, the efficient processing of massive quantities of data has become a difficult problem with respect to the IoV. To solve this problem, this paper proposed a task offloading strategy based on an improved GA for the IoV. An adaptive dynamic weight calculation method is proposed which transforms the optimization of task delay and energy consumption into the optimization of task overhead. A communication model and calculation model are constructed, and task priority is divided via the tomographic analysis method. Then, the GA is improved via the adaptive adjustment of the genetic operator, and it is subsequently used to solve the objective function of maximizing the system utility function, so as to obtain the optimal task offloading scheme. Compared with other strategies, this strategy is less affected by uneven loads of the MEC servers. In order to obtain the maximum system utility function value, the proposed strategy offloads vehicle-mounted safety computing tasks to the lightly-loaded MEC server during the offloading process. It retains some other computing tasks locally for processing; thus, the proposed strategy can successfully handle more in-vehicle safety computing tasks.

MEC services are not free. Thus, in subsequent research, we will combine the service capabilities, service costs, and task characteristics of MEC to formulate an offloading strategy that can interact with tasks in a timely manner and reduce tariffs.

**Author Contributions:** Conceptualization, methodology, software, validation, visualization, and writing—original draft, K.W.; project administration, resources, and writing—review and editing, X.W.; resources, writing—review and editing, and conceptualization, X.L. All authors have read and agreed to the published version of the manuscript.

**Funding:** This work was supported by the Fundamental Research Program of Shanxi Province (202103021223029); Scientific and Technological Innovation Programs of Higher Education Institutions in Shanxi, 2021 (STIP).

**Institutional Review Board Statement:** Not applicable.

**Informed Consent Statement:** Not applicable.

**Data Availability Statement:** The data used to support the findings of this study are included within the article.

**Conflicts of Interest:** The authors declare that there are no conflict of interest regarding the publication of this paper.

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
