# Peer review of "Sustainable Internet of Vehicles System: A Task Offloading Strategy Based on Improved Genetic Algorithm"

_sustainability, doi:10.3390/su15097506_

Round 1

Reviewer 1 Report (Previous Reviewer 2)

The presentation of the article is inconsistent with a third round. There are several oversights and a lack of rigor in this work. See, for example, the definition of the acronym GA in the abstract; the added sentence on lines 31-33; isolated subtitles and captions (lines 98 and 335); current equation (22); algorithm 1; Table 1; and many more.

 Extensive editing of the English language is required.

Author Response

Comments and Suggestions for Authors

The presentation of the article is inconsistent with a third round. There are several oversights and a lack of rigor in this work. See, for example, the definition of the acronym GA in the abstract; the added sentence on lines 31-33; isolated subtitles and captions (lines 98 and 335); current equation (22); algorithm 1; Table 1; and many more.

Comments on the Quality of English Language

 Extensive editing of the English language is required.

Response: Thank you for your review again. We have corrected several errors and improved the language.

Reviewer 2 Report (Previous Reviewer 5)

Many thanks for the efforts of the authors. I have no further comments.

The writing could be further polished.

Author Response

Comments and Suggestions for Authors

Many thanks for the efforts of the authors. I have no further comments.

Comments on the Quality of English Language

The writing could be further polished.

Response: Thank you for your review again. We have corrected several errors and improved the language.

Reviewer 3 Report (New Reviewer)

Summary/Contribution: The work proposes a task offloading method for optimizing resource allocation in the Internet of Vehicles (IoV) using an improved genetic algorithm (GA). The method maximizes the system utility function by dividing the system into communication and calculation models, and the improved GA is used to obtain an optimal task offloading scheme. The method addresses the issue of local optima by introducing Halton sequence for uniformly initial population distribution and adapting the GA for problem model and global optimal solution guarantee. Empirical study validates the proposed method, showing successful task offloading with a rate of task calculation completion of around 75%.

Suggestions/Comments:

1. Abstract: 

  a. One idea to improve this abstract is to provide more context on the specific challenges or problems that the proposed method is addressing. For example, the abstract could briefly describe the existing methods for task offloading in IoV networks, and explain the limitations or shortcomings of these methods that the proposed method seeks to overcome. This would help to better situate the research in the wider context of IoV network optimization and highlight the novelty and significance of the proposed approach.

b. Additionally, the abstract could provide more detail on the empirical study, such as the specific metrics used to evaluate the performance of the proposed method and any insights or implications that can be drawn from the results.

2. Introduction: Please outline the challenges that need to be addressed in implementing task offloading in MEC, specifically in terms of vehicle mobility management, computing resource allocation, and task offloading decision-making under limited computing resources.

3. Section 2: It will be useful to summarize the main findings of this section in tabular form in order to emphasize the limitations of existing works and the originality of the proposed approach.

4. The authors may add a paragraph that deals with the use of well-known formal methods for the protection of IoT systems from security breaches   5. For this purpose, they may include the following interesting papers (and others): 
a. https://link.springer.com/chapter/10.1007/978-3-030-13705-2_26 b. https://link.springer.com/chapter/10.1007/978-3-030-02807-7_9   6. Please explain the two cases of task offloading and how they are defined in the context of MEC servers, as well as the main components of the total delay associated with computing tasks that are offloaded to MEC servers.

7. About GA algorithms:   a. What is the theory behind the GA algorithm, and how does it work to generate a solution to a problem? Can you explain the role of genetic operators and fitness function in the GA algorithm?

b. How does the fitness function determine which individuals are eliminated from the population?   c. What are some advantages of using the GA algorithm over other optimization algorithms?   8. Please describe how the proposed task offloading strategy based on an improved GA for the IoV can be extended to handle other types of computing tasks besides in-vehicle safety computing tasks.

9. Can you propose modifications to the proposed task offloading strategy based on an improved GA for the IoV that would improve its efficiency in processing massive data?

10. How can the communication model and calculation model used in the proposed task offloading strategy based on an improved GA for the IoV be adapted to support real-time decision-making for traffic management purposes?

Acceptable.

Author Response

Comments and Suggestions for Authors

Summary/Contribution: The work proposes a task offloading method for optimizing resource allocation in the Internet of Vehicles (IoV) using an improved genetic algorithm (GA). The method maximizes the system utility function by dividing the system into communication and calculation models, and the improved GA is used to obtain an optimal task offloading scheme. The method addresses the issue of local optima by introducing Halton sequence for uniformly initial population distribution and adapting the GA for problem model and global optimal solution guarantee. Empirical study validates the proposed method, showing successful task offloading with a rate of task calculation completion of around 75%.

Suggestions/Comments:

1. Abstract: 

  1. One idea to improve this abstract is to provide more context on the specific challenges or problems that the proposed method is addressing. For example, the abstract could briefly describe the existing methods for task offloading in IoV networks, and explain the limitations or shortcomings of these methods that the proposed method seeks to overcome. This would help to better situate the research in the wider context of IoV network optimization and highlight the novelty and significance of the proposed approach.
    Additionally, the abstract could provide more detail on the empirical study, such as the specific metrics used to evaluate the performance of the proposed method and any insights or implications that can be drawn from the results.
    Response: We have supplemented the abstract according to your request.

  2. Introduction: Please outline the challenges that need to be addressed in implementing task offloading in MEC, specifically in terms of vehicle mobility management, computing resource allocation, and task offloading decision-making under limited computing resources.
    Response: The introduction already includes these contents:”The current MEC offloading strategies are all concentrated on small mobile terminals, which have limited energy consumption, slow movement, and often do not span the service scope of a single MEC in a short time...”

  3. Section 2: It will be useful to summarize the main findings of this section in tabular form in order to emphasize the limitations of existing works and the originality of the proposed approach.
    Response: Thank you for your suggestion. The limitations of the existing work and the originality of the proposed method have been summarized at the end of the introduction as requested by other reviewers, and there is no need to summarize them again in the form of a table.

  4. The authors may add a paragraph that deals with the use of well-known formal methods for the protection of IoT systems from security breaches   5. For this purpose, they may include the following interesting papers (and others): 
    a.https://link.springer.com/chapter/10.1007/978-3-030-13705-2_26 b. https://link.springer.com/chapter/10.1007/978-3-030-02807-7_9

Response: Thank you for your suggestion. After careful consideration, we believe that the introduction to formal methods for protecting IoT systems from security vulnerabilities is already covered, and this is not strongly related to the content of this study, so there is no need to set up a separate section. 

  1. Please explain the two cases of task offloading and how they are defined in the context of MEC servers, as well as the main components of the total delay associated with computing tasks that are offloaded to MEC servers.
    Response: The content you mentioned has been elaborated on in Section 3.

  2. About GA algorithms:  
  3. What is the theory behind the GA algorithm, and how does it work to generate a solution to a problem? Can you explain the role of genetic operators and fitness function in the GA algorithm?
    b. How does the fitness function determine which individuals are eliminated from the population?  
  4. What are some advantages of using the GA algorithm over other optimization algorithms?  

Response: The theory behind genetic algorithms has been elaborated in the article:”The theory of the GA surrounds an intelligent...The solution flow of the GA is shown in Fig. 2. ”. Moreover, genetic algorithm is a conventional algorithm that does not require further detailed explanation (some reviewers previously suggested deleting more introductions about GA). The role of genetic operator and fitness function in genetic algorithm is very basic, and it is also briefly described in this paper.

  1. Please describe how the proposed task offloading strategy based on an improved GA for the IoV can be extended to handle other types of computing tasks besides in-vehicle safety computing tasks.
    Response: So far, this study has not proposed to extend the proposed task offloading strategy based on improved IoV GA to handle other types of computing tasks besides vehicle safety computing tasks.

  2. Can you propose modifications to the proposed task offloading strategy based on an improved GA for the IoV that would improve its efficiency in processing massive data?
    Response: The idea you proposed is very forward-looking, and we will consider it in the future. MEC services are not free. Thus, in subsequent research, we will combine the service capabilities, service costs, and task characteristics of MEC to formulate an offloading strategy that can interact with tasks in a timely manner and reduce tariffs.

  3. How can the communication model and calculation model used in the proposed task offloading strategy based on an improved GA for the IoV be adapted to support real-time decision-making for traffic management purposes?

 Response: The real vehicle networking environment is extremely complex, with complex road conditions, different vehicle types, varying vehicle speeds, and diverse vehicle behaviors making corresponding modeling and research very difficult. For the convenience of research, this study first establishes a system model to model the actual complex road conditions. How to apply it to support real-time decision-making for traffic management purposes is a follow-up research topic.

Comments on the Quality of English Language

Acceptable.

Response: Thank you for your comment. We have improved the language.

Round 2

Reviewer 1 Report (Previous Reviewer 2)

All my concerns have been partially addressed.

Moderate editing of English language

Reviewer 3 Report (New Reviewer)

The authors considered all my comments and suggestions. Good luck.

Acceptable.

This manuscript is a resubmission of an earlier submission. The following is a list of the peer review reports and author responses from that submission.

Round 1

Reviewer 1 Report

The manuscript "Sustainable Internet of Vehicles System: A Task Unloading Strategy Based on Improved Genetic Algorithm" aims to add to the growing body of knowledge on smart transportation by proposing the use of a genetic algorithm. The topic is valuable and has a potential to be interesting to the readership of the journal. The paper is well structured and has many strongpoint. Accordingly, the paper is on a good way to be published. I would, nevertheless, recommend some improvements:

Major issues

#1 ABSTRACT: The abstract of the paper is technical. As the readership of the journal may not be experts in neither smart transportation nor genetic algorithms, I would suggest major improvements. Use simple economic or social intuition to re-write the abstract. It is important to explain why the results matter for readers not focused on your context. Any forward looking statements would motivate a comprehensive reading of this article 

#2 INTRODUCTION: Introduction of the paper mainly addresses the strongpoints of MEC compared to its older counterparts. Thus, the reader can be puzzled with what this paper is dealing with. First, explicitly state what is the aim of your study. Try to explain the main contributions of your paper as early as in the introduction of the paper.  

#3 In sections 2 and at the beginning of sub-section 4.3, after a thorough text similarity check, I detected inappropriate citation and heavy text similarity. I strongly advise additional improvements in the text.

Minor issues:

#1 Avoid using quotes for smart transportation. 

#2 In section 3.1. the use of bold letters is unnecessary. 

#3 In line 569, instead of using "The reference [17] used reinforcement...", use "Wang et al. [17] used reinforcement..." Similar is in line 99. This should be corrected throughout the text.

#4 I would advise the authors to cite other paper from the Sustainability journal that deal with similar topics.

#5 The authors should once again re-read the paper to correct minor issues. For instance, in the discussion part the authors claim "The proposed strategy will intelligently offload tasks when offloading tasks [...]". This seems quite recursive by nature.

Reviewer 2 Report

The article addresses a topic of current interest. However, the scientific rigor evidenced in the document is very low, showing deficiencies in multiple aspects. For these to be improved, the reviewer directs some suggestions and doubts to the authors.

In the current version of the document, the contributions of the research work seem limited and insufficient. Even in the description of these, a critical reading allows us to conclude that the second contribution described is contained in the first. In the reviewer's view, it has only been demonstrated that the proposal of this work outperforms the one presented in [17] and that for the scenarios used, the improved GA performance outperforms the performance of the traditional GA implementation.

In this sense, several questions arise: Why was only [17]'s proposal selected as a benchmark, when close to 15 closely related previous works were analyzed? Why was the benefit of the current proposal over previous work not clearly stated? Especially on own works such as [11] and the one published in the "Journal of Grid Computing" (not even cited).

Considering the complexity associated with IoV environments, other questions arise regarding simplifications: How valid is it to consider the distribution of RSUs uniformly? How valid is it to consider also the distribution of local tasks uniformly? What impact do these considerations have on the performance of the proposed algorithm? Is the use of the Halton sequence to generate the initial GA population beneficial for all IoV scenarios?

The research problem must be clear, it is suggested to describe an example where the most fundamental concepts are clear. In addition, the selection of performance metrics must be justified.

The reviewer suggests that references be used in blocks, without distinguishing their use before or after, see cases [1][2] and [7][8].

The formulation and presentation of the equations are poor, they are not properly centered, nor are punctuation marks used in them, their numbering is not vertically centered, and the margins of the next line to an equation are not used distinctively.

The quality of the figures is questionable, but Fig. 2 is even unnecessary.

Algorithm 1 is not properly presented and explained.

In black and white versions, in Figure 6 it is difficult to identify each strategy.

The need for the Discussion Section is questionable, in my opinion, these aspects could be dealt with in Section 5.

The conclusions do not contain conclusive and generalizable ideas, they seem more like an account of what was done.

Reviewer 3 Report

The selected topic is sound, but to proceed with publication, it requires a major revision. Some of the main highlights are given below:

1) The abstract is not clear at all. The authors should compile it throughout aiming to make it more catchy. 

2) Big concern: The introduction section described the previous work but there is no discussion of the author's contributions. The abstract is not in line with the introduction. The contribution is not highlighted.  

3)  In the related work, the authors should describe a relationship between the previous work and their contributions.

4) Why too many symbols are used? Many of them are unused or used just one time. 

5) Why section 3.1 is bold?

6) All the symbols in the text are presented unprofessionally. Same comments for all Equations.

7) Most of the Equations are drawn without discussing their parameters.

8) The caption of figure 2 is not clear. 

9)  Algorithm I has many parameters that are not discussed. The authors should revise it to make it clearer. Secondly, the authors should discuss it in detail. 

10) Table 2 should be right justified.

11) The lines of all graphs should be colored aiming to easily differentiate.

12) Why local unload in figure 4 zero on all inputs? explain it. 

13) The results are not clear. The authors should revise it entirely. 

Reviewer 4 Report

Broad comments. The authors have made a concise overview of the topic and a brief reference to existing literature. In general, the text is very well structured and has clearly defined topics. Some comments for improvement: he

1. The authors are encouraged to refine the introduction such that it provides information on the specific research questions that the work focuses on. The last part of section 2 could be for example transferred to section 1 and refined in a way to describe not only what this work does but what this work tries to research.

2. Besides the main novelty of the work and the impact of the outcome should be better-emphasized Authors are encouraged to point out the key message and the potential benefits of their work combined with the justification of the importance of the problem.

3. Authors could consider enhancing section 2 such that the literature review could cover recent approaches in traffic/speed monitoring using innovative technologies that provide computing at the edge.

[1] C. Spandonidis, F. Giannopoulos, E. Sedikos, D. Reppas and P. Theodoropoulos, "Development of a MEMS-Based IoV System for Augmenting Road Traffic Surey," in IEEE Transactions on Instrumentation and Measurement, vol. 71, pp. 1-8, 2022, Art no. 9510908, doi: 10.1109/TIM.2022.3198755.

[2] S. Wan, S. Ding, and C. Chen, “Edge computing enabled video segmentation for real-time traffic monitoring in internet of vehicles,” Pattern Recognit., vol. 121, p. 108146, Jan. 2022, doi: 10.1016/j.patcog.2021.108146.

[3] F. Busacca, C. Grasso, S. Palazzo, and G. Schembra, “A Smart Road Side Unit in a Microeolic Box to Provide Edge Computing for Vehicular Applications,” IEEE Trans. Green Commun. Netw., pp. 1–1, 2022, doi: 10.1109/TGCN.2022.3187674.

4. Information provided in figure 1 is not clear. Authors could consider enhancing the figure by adding information (i.e. for connectivity protocols etc).

5. A flowchart could be added in parallel to algorithm 1, to better illustrate the flow of the process.

6. The authors are encouraged to enhance the relevant section with a discussion regarding the limitations and restrictions they see towards the direction of Smart retail high streets.

7. The authors provided a very nice and well-written work. Conclusions should highlight the work done and give a summary of both the novelty and the impact of the work.

Reviewer 5 Report

In the manuscript, the authors have proposed an improved Genetic Algorithm (GA)-based task offloading strategy for the Internet of Vehicles (IoV) systems. The proposed improved GA is presented in detail and analyzed with simulation works. However, some concerns still need to be addressed before the acceptance.

  1. The writing is horrible and hard to follow. Some sentences are not readable at all. Serious proofreading with a native speaker is highly recommended.
  2. Is there any difference between task offloading and task unloading? The authors mixedly use these two terms in the manuscript.
  3. In (4), $d_1$ refers to the distance between the vehicle and RSU. Afterward, $d_l$ is defined as a time-variable distance between the vehicle and the center of the RSU coverage area in (5). These two notations are separated and listed in Table 1 as well. What is the difference between these two notations?
  4. In (5), $d_{\Delta}$ refers to the distance between the vehicle driving level and RSU. What does “vehicle driving level” mean?
  5. In (8), the time cost of data transmission is calculated by $D_i/v_{ij}$, where $D_i$ is the data size, and $v_{ij}$ is the distance between the vehicle and MEC server $j$. It does not make sense. Should be the time cost of data transmission determined by data size over data rate instead of distance?
  6. Could the authors please explain the expression of $a_{jk}=1/a_{kj}=n$ in (14)? Why does $n$ count from 1 to 9?
  7. In (19), the system utility function is defined by $P=\sum_{i=1}^N\omega_k/\theta$. There is no variable $i$ in the expression. How can it sum from i=1 to N?
  8. In (20), $y_i$ in the first constraint refers to the decision of local computing or offloading to MEC for computing. However, it is not involved in the objective function at all. Could the authors please check the expressions?
  9. Fig. 2 depicts the comparison of the traditional algorithm and the GA algorithm. First, the sub-caption of the GA algorithm seems missing. Moreover, GA is a classical algorithm. Providing the low quality of Fig. 2, is it necessary to provide Fig. 2 there?
  10. In (22), the expressions of child 1 and child 2 are exactly the same.

Round 2

Reviewer 2 Report

The main concerns of this reviewer have not been addressed, nor has the paper been worked on to prevent them from being the concerns of future readers. Regarding the first four comments, the authors did not try to avoid the deficiencies, discussing the issues in depth and adding related comments in the new version of the manuscript. Rather, the authors used the words of the reviewer to justify that no change would be made.

Author Response

The reply is in the coverletter.

Reviewer 3 Report

All my comments are addressed.

Reviewer 5 Report

Many thanks for the efforts of the authors. Some of my concerns have been addressed. However, some are still there.

  1. In (4), $d_1$ refers to the distance between the vehicle and RSU. Afterward, $d_l$ is defined as a time-variable distance between the vehicle and the center of the RSU coverage area in (5). It is highly suggested that the authors keep the consistency of the description. For example, the authors can add an explanation that the distance between the vehicle and RSU refers to the distance between the vehicle and the center of the RSU coverage area. Otherwise, the readers may get confused. The issue is not well addressed by deleting the Table of Notations.
  2. The explanation of the vehicle driving level is suggested to be added to the manuscript.
  3. In(22), both $\phi_(child1)$ and $\phi_(child2)$ equal to $(1-\epsilon)\phi(parent1)+\epsilon\phi(parent2)$. The only difference is that the additive order is commutative. Due to the additive commutative law, I believe they are the same. Could the authors please kindly tell the difference?